# Effect of SRB and Applied Potential on Stress Corrosion Behavior of X80 Steel in High-pH Soil Simulated Solution

**DOI:** 10.3390/ma14226981

**Published:** 2021-11-18

**Authors:** Congmin Xu, Haoran Gao, Wensheng Zhu, Wenyuan Wang, Can Sun, Yueqing Chen

**Affiliations:** 1School of Materials Science and Engineering, Xi’an Shiyou University, Xi’an 710065, China; 19211050483@stumail.xsyu.edu.cn (H.G.); 182090409@stumail.xsyu.edu.cn (W.W.); ssccwenzhang@163.com (C.S.); 19212050513@stumail.xsyu.edu.cn (Y.C.); 2CNOOC Changzhou Paint and Coating Industry Research Institute Co., Ltd., Changzhou 213000, China; zhuwsh@cnooc.com.cn

**Keywords:** X80 pipeline steel, high-pH soil simulated solution, stress corrosion sensitivity, cathodic protection potential, sulfate-reducing bacteria

## Abstract

The effect of SRB and applied potential on the stress corrosion sensitivity of X80 pipeline steel was analyzed in high-pH soil simulated solution under different conditions using a slow strain rate tensile test, electrochemical test, and electronic microanalysis. The experimental results showed that X80 pipeline steel has a certain degree of SCC sensitivity in high-pH simulated solution, and the crack growth mode was trans-granular stress corrosion cracking. In a sterile environment, the SCC mechanism of X80 steel was a mixture mechanism of anode dissolution and hydrogen embrittlement at −850 mV potential, while X80 steel had the lowest SCC sensitivity due to the weak effect of AD and HE; after Sulfate Reducing Bacteria (SRB) were inoculated, the SCC mechanism of X80 steel was an AD–membrane rupture mechanism at −850 mV potential. The synergistic effect of Cl^−^ and SRB formed an oxygen concentration cell and an acidification microenvironment in the pitting corrosion pit, and this promoted the formation of pitting corrosion which induced crack nucleation, thus significantly improving the SCC sensitivity of X80 steel. The strong cathodic polarization promoted the local corrosion caused by SRB metabolism in the presence of bacteria, whereby the SCC sensitivity in the presence of bacteria was higher than that in sterile conditions under strong cathodic potential.

## 1. Introduction

Approximately 80% of the pipelines transporting oil and gas are buried underground. X80 pipeline steel is a kind of low-carbon, micro-alloyed high-grade steel, used widely for building buried gas transmission pipelines [1]. However, these underground pipelines are affected by soil corrosion around the world. Cathodic protection (CP) is a technique used to protect pipeline steels from corrosion attack, by shifting their surface potential to safe ranges [2,3]. Once the protective coating of pipelines is damaged, it inevitably results in microbiologically induced corrosion (MIC) of the pipeline in the soil environment, with sulfate-reducing bacteria (SRB) representing the most critical corrosive bacteria among numerous microorganisms [4].

Stress corrosion cracking (SCC) has been recognized as one of the fatal threats to pipeline transportation safety, which has strong and fast destructive power and often occurs suddenly without obvious warning [5,6]. Current research shows that stress corrosion causing pipeline rupture is one of the main failure modes [7,8]. Fan et al. [9] found that SCC showed different mechanisms at different applied potentials, including anodic dissolution (AD), hydrogen embrittlement (HE), and a mixed mechanism. Some scholars believe that microorganisms can cause hydrogen embrittlement (HE), which makes metals crack under the synergistic action of stress and hydrogen, putting forward the concept of “microbe-assisted cracking (MAC)” [10,11].

In recent years, a large number of microorganisms, especially anaerobic, have been found under stripping coatings pipes [12]. SCC is caused by pitting pits formed during local corrosion. Therefore, it is crucial to study the role of SRB and other microorganisms in SCC. Field failure analysis shows that SRB or other microorganisms can promote SCC [13,14,15]. Wu et al. [16] studied the behavior of X80 pipeline steel subjected to stress and SRB, which induced pitting corrosion. The pitting corrosion on the bottom was due to a mechanical–electrochemical effect, which accelerated crack nucleation and propagation. Another study found [17] that SRB promote the SCC sensitivity of X80 steel in the soil, and SRB in the rapid cell growth period had the most significant effect on the sensitivity to SCC. In addition, the SCC sensitivity of X80 steel is promoted by the collective effect of cathodic protection potential and SRB, whereas, with a negative shift of the cathodic protection potential, this synergistic effect is reduced and the SCC sensitivity is decreased. Wang et al. [18] found that, when the applied protective potential is −1275 mV, the activity of SRB intensifies the cathodic hydrogen evolution of X80 steel, and the SCC mechanism of X80 steel in an acidic soil solution at this time is hydrogen-induced cracking. Wang et al. [19] found that, due to the metabolic activity of SRB and the cathode plan, the current density of X80 in the solution increased sharply to three times that of no SRB, and severe hydrogen bubbles appeared on the surface of the sample under the action of SRB, which induced severe hydrogen-induced cracking in the pipeline. Wang et al. [20] studied the SCC behavior of X80 pipeline steel in an acidic soil simulation solution and found that the presence of SRB promotes SCC susceptibility. When the number of SRB increased, X80 steel was more prone to stress corrosion. When the number of SRB reached its maximum value in the day, the corrosion was most serious.

Due to the influence of the soil and SRB, applying cathodic protection potential to the buried pipeline may reduce the pipeline potential, resulting in an increase in SCC sensitivity. Generally, the cathodic protection potential cannot be stabilized at an appropriate value [21,22]. Therefore, it is necessary to investigate the SCC cathodic protection of pipeline steel in an SRB soil environment.

In this paper, the influence of SRB and applied potential on the SCC sensitivity of X80 steel in high-pH soil simulated solution is studied, and the corrosion behavior of X80 steel is investigated in order to provide theoretical guidance for practical engineering.

## 2. Materials and Methods

### 2.1. Experimental Preparation

X80 pipeline steel was used in this experiment with the following chemical composition (weight percent, wt.%): C 0.047, Mn 1.81, Si 0.19, P 0.01, S 0.0021, Cr 0.35, Mo 0.11, Nb 0.066, Ni 0.14, V 0.003, Ti 0.015, Cu 0.17, and Fe balance. Mechanical properties at room temperature were a yield strength (σ_0.5_) of 604 MPa, tensile strength (σ_b_) of 727 MPa, and elongation after fracture of 38%.

The SRB used in this experiment were derived from the soil in southwest China, obtained through enrichment, separation, and purification. SRB were cultured in standard SRB culture medium [23,24], consisting of NH_4_Cl 1.0 g, CaCl_2_·2H_2_O 0.1 g, K_2_HPO_4_ 0.5 g, Na_2_SO_4_ 2 g, MgSO_4_·7H_2_O 2.0 g, yeast paste 1.0 g, and 60% sodium lactate solution 6 mL. The reagents were dissolved in 1000 mL of water, the pH was adjusted to 7.2 ± 0.2 using NaOH or HCl solution, and then the SRB culture medium was autoclaved at 12 °C for 20 min.

The solution used in this experiment was a high-pH soil simulated solution with a composition of 0.5 mol/L Na_2_CO_3_ + 1 mol/L NaHCO_3_ + SRB culture medium, adjusted to pH 9.62 using NaOH solution.

Plate specimens shown in Figure 1 were used for the slow strain rate tensile (SSRT) test. The specimens were polished to 1000 grit using SiC sandpaper; then, they were cleaned with ethanol and degreased with acetone to remove surface grease and debris.

### 2.2. SSRT

A MFDL100 SSRT (slow strain rate test) machine was used for the tensile test. The strain rate was 5 × 10^−7^ s^−1^; the test temperature was room temperature. Three cathodic protection potentials were selected: −850 mV, −1000 mV, and −1200 mV r vs. SCE. A three-electrode system was used for the applied potential test. After the test, the specimen fracture surface was observed using a JSM-6390 (JEOL Ltd., Tokyo, Japan) scanning electron microscope (SEM) and analyzed by EDS.

The elongation at break or shrinkage reflects the toughness and brittleness of the material. A greater elongation at break or shrinkage denotes greater toughness. The tensile strength is related to the degree of corrosion of the medium. A greater tensile strength results in a smaller degree of corrosion, a longer life of the steel, and a longer fracture time. Tensile strength, elongation at break, and cross-sectional reduction rate reflect the single mechanical properties of SCC. In order to better reflect the stress corrosion sensitivity, the stress corrosion degree of steel is evaluated according to the tensile strength and elongation at break of the steel in corrosive and inert media, and the stress corrosion sensitivity factor is defined as follows:(1)$$I_{SSRT}=1-\frac{\sigma_{fw}\cdot \left(1+\delta_{fw}\right)}{\sigma_{fA}\cdot \left(1+\delta_{fA}\right)}$$
where $\sigma_{fw}$ is fracture strength of the specimen in environmental medium (MPa), $\sigma_{fA}$ is the fracture strength of specimens in inert medium (in air) (MPa), $\delta_{fw}$ is the specimen elongation after fracture in environmental medium (%), $\delta_{fA}$ is the specimen elongation after fracture in inert medium (in air), $I_{SSRT}$ is the stress corrosion sensitivity factor of the material–medium system, and the SCC sensitivity increases gradually from 0–1.

### 2.3. Electrochemical Testing

The electrochemical tests were carried out using an M2273 electrochemical workstation (EG&G, Gaithersburg, MD, USA) with a three-electrode system. Working electrode potentials of X80 steel were referred to a saturated calomel electrode (SCE). The auxiliary electrode was a Pt plate. The electrochemical specimens of X80 steel were sterilized by UV for 30 min before use. The specimens were immersed in two solutions, which were sterile or inoculated with bacteria. Different cathode potentials (−850, −1000, and −1200 mV) were applied to the specimens. The polarization curves were determined potentiodynamically using a fast scan rate (50 mV/s) and slow scan rate (0.5 mV/s). The scanning range of potentiodynamic polarization was from −400 mV (vs. *E*_corr_) to +1600 mV.

## 3. Results

### 3.1. The Results of SSRT

Figure 2 shows the stress–strain curves of X80 steel in high-pH solution (pH = 9.62) under different conditions. In the sterile solution, the tensile strength of the X80 specimen at −850 mV was higher than that of the air-tension specimen, and the tensile strength at the remaining applied potentials was lower than that of the air-tension specimen. Moreover, with a negative shift of the applied potential, the tensile strength significantly decreased. After SRB was inoculated, the tensile strength and strain of the X80 specimen at different applied potentials were lower than those of the tensile specimen in air, indicating a certain SCC sensitivity of X80 steel in high-pH solution.

Figure 3 shows the broken line diagram of the *I_SSRT_* of X80 steel under different conditions in high-pH solution. In a sterile high-pH solution, the $I_{SSRT}$ value of X80 steel increased with a negative shift of potential, and the broken line presented an upward trend, indicating that the stress corrosion sensitivity increased continuously. After SRB were inoculated in high-pH solution, the $I_{SSRT}$ value of X80 steel showed an up–down–up trend with a negative shift of potential, exhibiting the highest SCC sensitivity at −1200 mV. It was found that the SCC sensitivity was significantly increased at −850 mV with SRB compared to without SRB, whereas it decreased under a strong cathode potential with SRB compared to without SRB. To sum up, the SCC sensitivity of X80 steel was ranked as *I* (at OCP) < *I* (at −850 mV) < *I* (at −1000 mV) < *I* (at −1200 mV) in the sterile environment. In the SRB environment, the order on SCC sensitivity of X80 steel was *I* (at OCP) < *I* (at −1000 mV) < *I* (at −850 mV) < *I* (at −1200 mV).

### 3.2. SEM Analysis of Fracture Morphology

Figure 4 shows the SEM morphology of the fracture cross-section and side view of X80 steel under different conditions in high-pH solution. The actual service environment of X80 pipeline steel is basically oxygen-free; thus, oxygen was removed during the experimental operation. The high-pH solution contained HCO_3_^−^, H^+^, and H_2_CO_3_ and the cathode reaction was as shown in Equations (2)–(5) [25].
2H^+^ + 2e^−^ → H_2_(2)
2H_2_CO_3_− + 2e^−^ → 2CO_3_^2−^ + H_2_(3)
2H_2_O + 2e^−^ → 2H_2_ + 2OH^−^(4)
2H_2_CO_3_ + 2e^−^ → H_2_ + 2HCO_3_^−^(5)

The anode region reacts as follows [24]:Fe → Fe^2+^ + 2e^−^(6)
Fe^2+^ + 2OH^−^ → Fe(OH)_2_(7)
Fe^2+^ + CO_3_^2−^ → FeCO_3_(8)

There were obvious dimples and several micropores in the tensile fracture cross-section of X80 steel in air, and serpentine slip was obviously reflected on the local dimple wall (Figure 4a), indicating ductile fracture features of dimples and micropores. In the sterile environment, dimples and micropores also existed in the cross-section of specimens at −850 mV and OCP, but the dimples and micropores became smaller and shallower (Figure 4b,d), whereas the fracture mode was ductile fracture; at −1000 mV potential, the fracture morphology was dimple + quasi cleavage mixed fracture accompanied by a small number of micropores, indicating that the fracture mode gradually changed from ductile fracture to brittle fracture starting from −1000 mV potential. The SCC sensitivity of X80 steel was further increased at −1200 mV potential, and the fluvial pattern + cleavage morphology appeared at the fracture surface, accompanied by the appearance of a tearing edge, while the brittle fracture characteristics of the specimen are obvious. When SRB were inoculated in high-pH solution, the dimples on the specimens show dense and deep pores with dislocation accumulation at OCP, showing typical ductile fracture characteristics; at −850 mV, −1000 mV, and −1200 mV potentials, the fracture surface presented quasi-cleavage fracture and a fluvial pattern, exhibiting more obvious brittle fracture characteristics than that under sterile condition.

It can be seen from the side view of the fracture surfaces that secondary cracks appeared in all the specimens except that in air, indicating that X80 steel has a certain SCC sensitivity in high-pH solution. The difference is that the secondary cracks on the fracture surface were not only deeper than those under sterile conditions, but also accompanied by more pitting corrosion in the SRB environment. However, the side surface of the specimen in the sterile environment was flat compared to that in the SRB environment, indicating that SRB promoted the formation of pitting corrosion and significantly improved the SCC sensitivity of X80 steel. It was also found from the side view of X80 steel fracture that the secondary cracks were mostly wide and deep, and the crack direction was perpendicular to the tensile stress direction and propagated along the straight line. Thus, the crack growth mode of X80 steel was trans-granular stress corrosion cracking (TGSCC) [26].

Figure 5 shows the SEM morphology and EDS analysis results of X80 steel immersed in the SRB environment for 3 days at −850 mV and −1000 mV potentials.

Due to the life activities of SRB, it can be observed that the side surface of the specimen was covered by corrosion products and metabolites of SRB. In the presence of SRB, the reaction of SO_4_^2−^ and H was promoted, thus producing S^2^ which then combined with H^+^ to produce a large amount of H_2_S. The chemical reaction was as follows [25]:(9)$${\mathrm{SO}}_{4}{}^{2-}+8H\;\to \;S2^{-}+4H_{2}O,$$
(10)$$S^{2-}+2H^{+}\;\to \;H_{2}S,$$
(11)$$\mathrm{Fe}+H_{2}S\;\to \;\mathrm{Fe}+{\mathrm{HS}}^{-}+H^{+}\;\to \;\mathrm{FeS}+H+H.$$

The flocculent corrosion products composed of SRB were unevenly distributed on the side surface of the specimen at −850 mV potential. According to EDS analysis, the corrosion products were mainly oxides with a small amount of sulfide. The existence of S indicates that the metabolic activity of SRB produced sulfide and precipitates into the biofilm. The flocculent corrosion products of SRB were obviously reduced, and the content of S decreased obviously when the potential was −1000 mV; the content of S was the lowest when the potential was −1200 mV, indicating that SRB growth is inhibited under −1000 mV and −1200 mV cathodic potential.

### 3.3. Polarization Curve

The polarization curves at different scan rates can be used to determine the metal SCC sensitivity in certain environments. The sample surface at the crack tip cannot be fully polarized or covered by a corrosion product film during fast scanning [27]. The rapid scanning polarization curve can be used to represent the electrochemical behavior of the metal surface at crack tip, whereas the slow scanning polarization curve represents the electrochemical behavior of the metal surface at the crack wall (non-crack tip). Therefore, the SCC crack mechanism under different additional point conditions can be judged according to the difference in the planning curve under different scanning rates, and the SCC sensitivity evaluation system can be established. In addition, the difference between *i*_s_ (slow scan current density) and *i*_q_ (quick scan current density) can be used to express the strength of the AD mechanism. A larger gap between *i*_s_ and *i*_q_ denotes a stronger action mechanism of AD [28]. Figure 6 shows the polarization curves of X80 steel in open-circuit potential (OCP) after being immersed in different solutions for 3 days. According to the difference in the zero-current potential between the fast scan polarization curve and slow scan polarization curve, the applied potential scan could be divided into three regions. Firstly, above the self-corrosion potential, *i*_s_ was always smaller than *i*_q_, which shows that the corrosion rate of the crack tip was faster than that of the crack wall, while the AD mechanism of SCC was dominant. In addition, there was an obvious activation–passivation transformation and a stable passivation area in the anode area, showing that the anode area at the crack tip was controlled by an active dissolution process at first and then passivated. Lastly, the rupture of the passive film led to further dissolution of the crack tip. According to these characteristics, the cathodic protection potential of −850 mV with SRB was in the potential region of the AD–membrane rupture mechanism. Secondly, between the self-corrosion potential and the quick-scan zero-current potential, the cathode hydrogen evolution reaction occurred at the crack wall (non-crack tip). With the continuous negative shift of potential, the gap between *i*_s_ and *i*_q_ was gradually reduced, indicating that the AD effect was gradually reduced. The SCC in this interval was dominated by an AD + HE mixed mechanism. According to these characteristics, the protection potential at −850 mV without SRB and the self-corrosion potential with SRB were both in this region. Thirdly, under the OCP, the fast scan and slow scan curves were relatively close; thus, the AD effect could be ignored. At the beginning of the cathode reaction, the hydrogen evolution reaction occurred, and a large amount of hydrogen was generated on the metal surface. According to these characteristics, the cathodic protection potential at −1000 mV and −1200 mV in both environments (with and without SRB) was in this region.

Figure 7 and Table 1 respectively show the polarization curves and Tafel fitting results of different cathodic protection potentials of X80 steel after being immersed in sterile and inoculated SRB environments for 3 days. The potentiodynamic polarization curves were determined with a scan rate of 0.5 mV/s and a scan range of −400 mV (vs. *E*_corr_) to +1600 mV. In Table 1, *b*_a_ denotes the anodic Tafel slope, *b*_c_ denotes the cathodic Tafel slope, *E*_corr_ is the self-corrosion potential, and *i*_corr_ is the self-corrosion current density. At −850 mV potential, *i*_corr_ and *b*_a_ in the sterile environment were significantly lower than in the inoculated SRB environment, indicating that SRB significantly promoted the anodic dissolution of X80 steel, intensified the corrosion, induced pitting, and then caused more pitting pits on the metal surface (Figure 4e), which significantly increased the SCC sensitivity of X80 steel. At −1000 mV potential, *i*_corr_ and *b*_a_ in the sterile environment were higher than in the inoculation SRB environment, indicating that the AD mechanism was weak, and the strong cathode potential inhibited the local corrosion caused by SRB metabolism, which greatly reduced the SCC sensitivity of X80 steel.

In sterile solutions, *i*_corr_ greatly increased with the negative shift of applied potential. However, in inoculated SRB solution, *i*_corr_ had the minimum value at −1000 mV potential with respect to −850 mV and −1200 mV potentials, indicating that the −1000 mV potential had a good cathodic protection effect and significantly inhibited SRB growth, which greatly reduced the corrosion rate of X80 steel, but both −850 mV (the poorly protective) and −1200 mV (the overprotective) potentials could enhance the SCC probability. The Tafel fitting results of polarization curves of X80 steel indicated that the applied potentials affected the X80 electrode surface state, and the metabolic activity of SRB in solution significantly accelerated the corrosion rate of X80 steel.

## 4. Discussion

The current research shows that SCC propagation can be divided into crack nucleation first, followed by the initiation stage and then the crack propagation stage [29]. In a sterile environment, at −850 mV, an AD + HE mixed mechanism was present, cathode polarization inhibited anodic dissolution, and the crack tip was accompanied by an anodic reaction. The two mechanisms produced relatively low effects, leading to a low SCC sensitivity at −850 mV. At −1000 mV and −1200 mV, the specimen showed significant brittleness which was due to atomic hydrogen generated by the cathodic reaction entering the metal surface under the strong cathode potential, which reduced the toughness of the steel and accelerated the crack propagation, thus promoting the occurrence of HIC. The brittle fracture characteristics on the X80 steel fracture surface were more obvious (Figure 4f,h).

In the presence of SRB, the reaction of SO_4_^2−^ and H was promoted, producing S^2−^ which then combined with H^+^ to produce a large amount of H_2_S. The chemical reaction is shown in Equations (9)–(11).

At −850 mV, it can be seen that SRB was attached to the metal surface and produced a biofilm via SRB metabolic reaction, which was not uniformly distributed on the surface of X80 steel, thus resulting in an oxygen concentration gradient on the metal surface (Figure 5a,b). In addition, SRB metabolically generated extracellular polymeric substances (EPS), which had a strong complexing effect on Fe^2+^ and promoted the anodic dissolution of the metal. Therefore, in this region, local corrosion was more likely to occur. On the other hand, the highly corrosive Cl^−^ contained in the solution combined with H^+^ to form HCl. The synergistic effect on Cl^−^ and SRB was more likely to promote the generation of pitting corrosion, which induced crack nucleation and promoted crack propagation, thus significantly improving the SCC sensitivity of X80 steel. At −1000 mV, due to the strong cathode potential, the biofilm and EPS produced by SRB metabolism were inhibited, and the SCC sensitivity of X80 steel was the minimum in the inoculated SRB environment. At the −1200 mV strong cathode potential, a large amount of hydrogen was produced on the metal surface, while the sulfide produced by the SRB metabolic activities also hindered the conversion process of hydrogen atoms to hydrogen molecules; therefore, more hydrogen atoms permeated into the metal matrix, increasing the sensitivity of steel to HE and greatly improving the SCC sensitivity of X80 steel.

## 5. Conclusions

(1)X80 pipeline steel showed SCC sensitivity in a high-pH simulated solution, and the crack growth mode was TGSCC.(2)In a sterile environment, the SCC sensitivity of X80 steel greatly increased with a negative shift of applied potential. At −850 mV, the SCC mechanism was AD + HE. However, in inoculated SRB solution, the SCC sensitivity of X80 steel was the minimum at −1000 mV potential, indicating that −1000 mV had a good cathodic protection effect and significantly inhibited SRB growth. At −850 mV with SRB, the SCC mechanism was AD–membrane rupture mechanism. At −1000 mV and −1200 mV with and without SRB, the SCC mechanism was controlled by HE.(3)The applied cathodic potentials affected the X80 electrode surface state, and the metabolic activity of SRB significantly accelerated the corrosion rate of X80 steel. Poorly protective and overprotective potentials could enhance the probability of SCC.(4)The sulfide produced by SRB also promoted hydrogen penetration into the steel matrix, thus increasing the steel’s HE sensitivity. The synergistic effect of Cl^−^ and SRB formed an oxygen concentration cell and an acidification microenvironment, which promoted the formation of pitting and induced crack nucleation, thus improving the SCC sensitivity of X80 steel.

## Figures and Tables

**Figure 1 materials-14-06981-f001:**
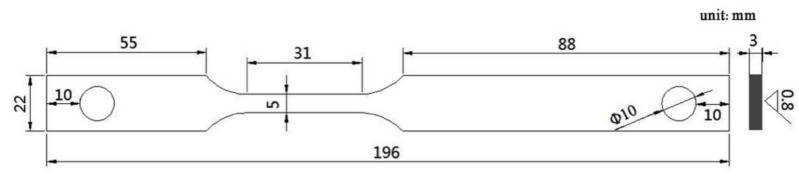
Drawing of the tensile specimen with slow strain rate.

**Figure 2 materials-14-06981-f002:**
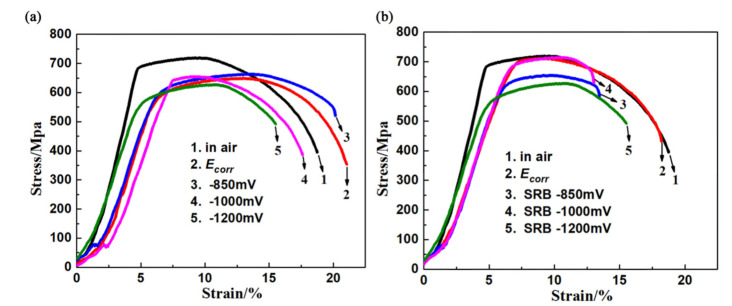
Engineering stress–strain curves of X80 steel in high-pH solution under different conditions: (**a**) without SRB; (**b**) with SRB.

**Figure 3 materials-14-06981-f003:**
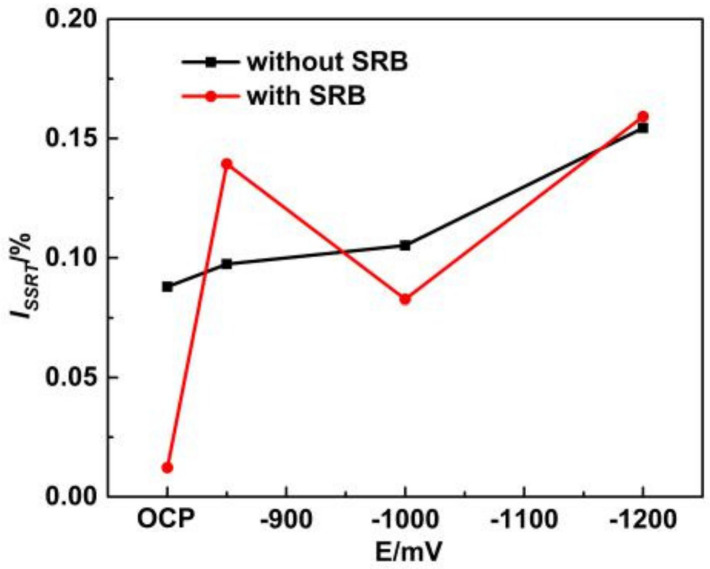
Broken line diagram of *I_SSRT_* of X80 steel under different conditions in high-pH solution.

**Figure 4 materials-14-06981-f004:**
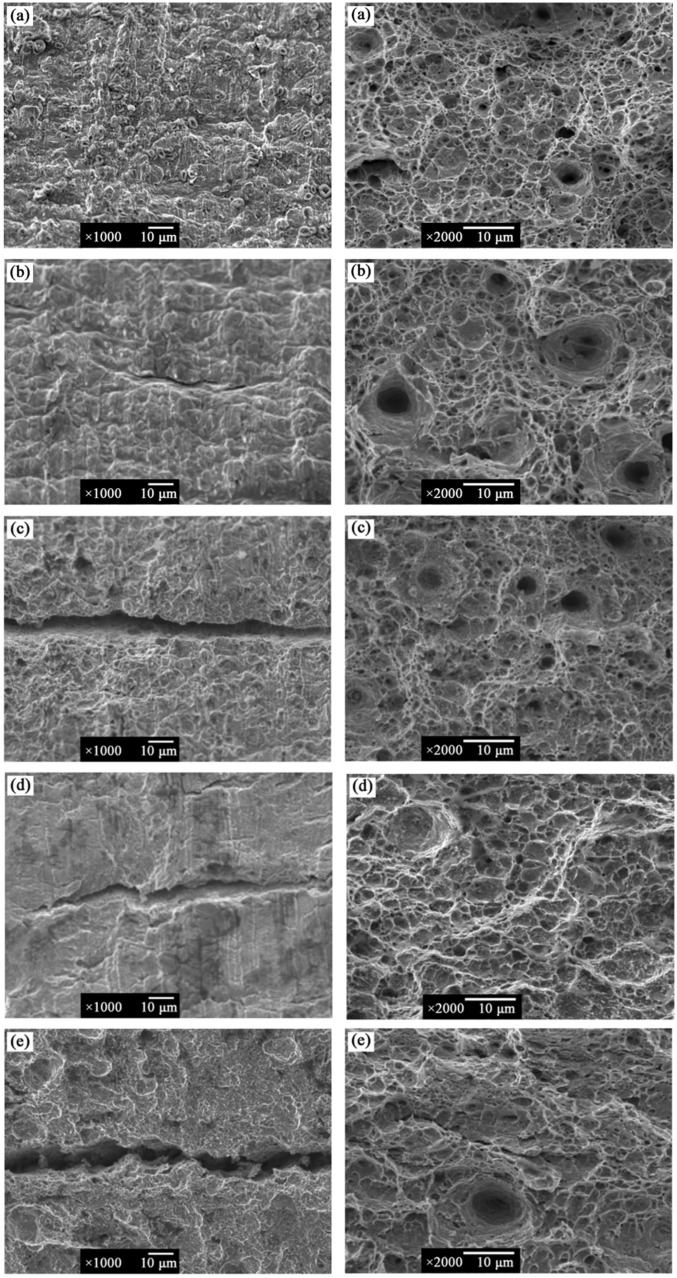
SEM images of X80 steel fracture surface under different conditions in high-pH solution: (**a**) in air; (**b**) without SRB at OCP; (**c**) with SRB at OCP; (**d**) without SRB at −850 mV; (**e**) with SRB at −850 mV; (**f**) without SRB at −1000 mV; (**g**) with SRB at −1000 mV; (**h**) without SRB at −1200 mV; (**i**) with SRB at −1200 mV.

**Figure 5 materials-14-06981-f005:**
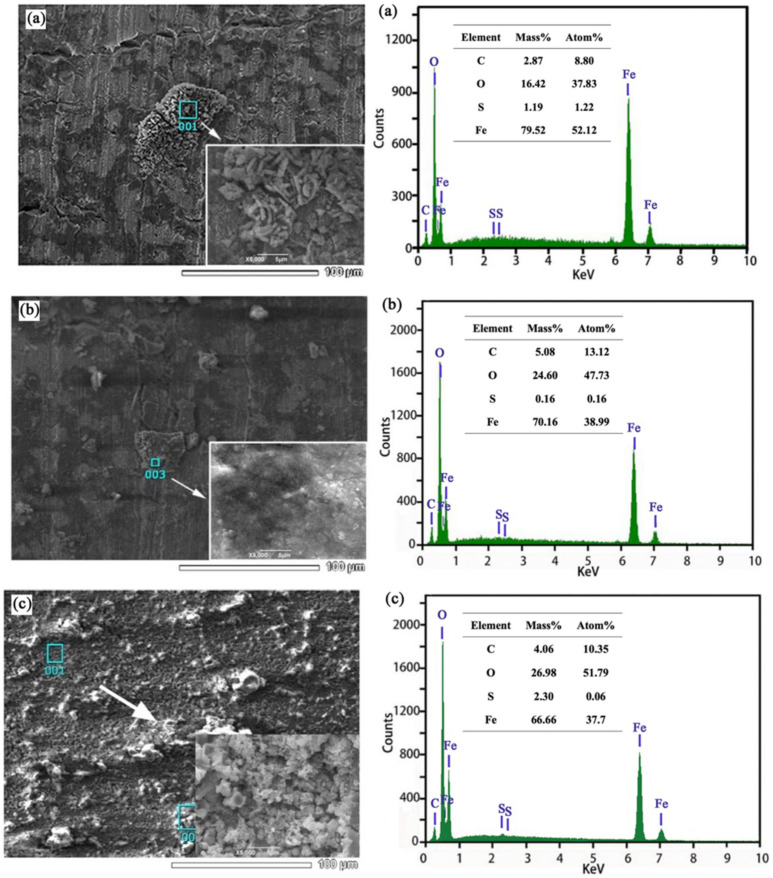
SEM morphology and EDS analysis of X80 steel immersed in SRB environment for 3 days: (**a**) at −850 mV; (**b**) at −1000 mV; (**c**) at −1200 mV.

**Figure 6 materials-14-06981-f006:**
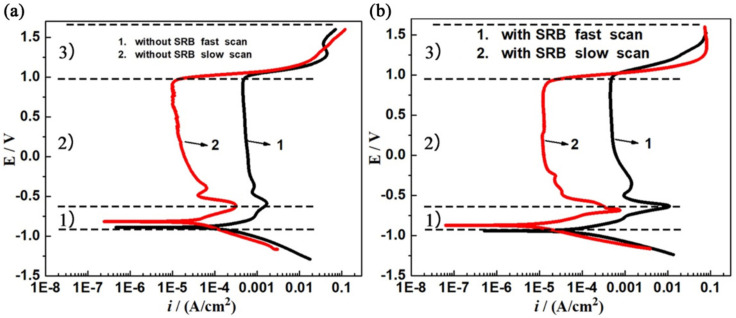
Fast (50 mV/s) and slow (0.5 mV/s) scan polarization curves of X80 steel under different conditions: (**a**) without SRB, (**b**) with SRB.

**Figure 7 materials-14-06981-f007:**
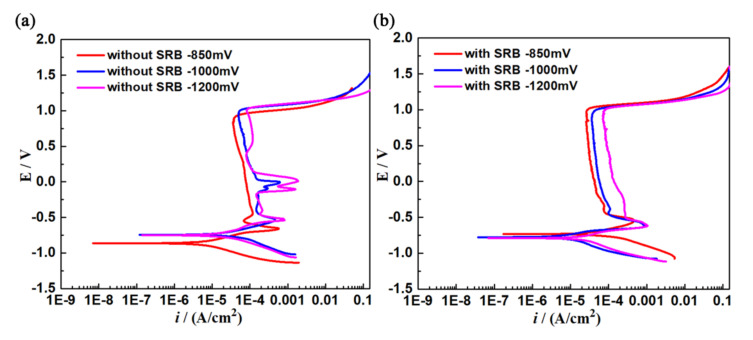
Polarization curves (0.5 mV/s) of different cathodic protection potentials of X80 steel in sterile and inoculated environments for 3 days: (**a**) without SRB; (**b**) with SRB.

**Table 1 materials-14-06981-t001:** Tafel fitting data of polarization curves of X80 steel in sterile and inoculated environments.

	Parameter	*E* (mV)	*b*_a_ (mV)	*b*_c_ (mV)	*i*_corr_ (A/cm^2^)	*E*_corr_ (mV)
Environmental						
Without SRB		−850 mV	126.6	153.1	0.58 × 10^−5^	−863
		−1000 mV	215.9	162.9	3.13 × 10^−5^	−744
		−1200 mV	195.7	167.9	3.41 × 10^−5^	−749
With SRB		−850 mV	211.96	133	3.24 × 10^−5^	−733
		−1000 mV	278.5	213.4	1.73 × 10^−5^	−828
		−1200 mV	211.3	174.3	3.48 × 10^−5^	−789

## Data Availability

Data sharing not applicable. No new data were created or analyzed in this study. Data sharing is not applicable to this article.

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
