# Peer review of "Effect of SRB and Applied Potential on Stress Corrosion Behavior of X80 Steel in High-pH Soil Simulated Solution"

_materials, 2021, doi:10.3390/ma14226981_

Round 1
Reviewer 1 Report
This manuscript studied the synergetic effect of sulfate-reducing bacteria under cathodic polarization on the stress corrosion cracking behavior of an X80 steel in near-neutral pH soil simulated solution. The presented manuscript is interesting.
Below you will find my comments.
Comments
Page 3: You forgot to report the EIS measurements in the manuscript. Lines 115-118
Page 3: What do you mean high pH solution? You have only one pH. (7.2), as reported on line 82. Line 121
Page 5: SEM images of the crack wall must be enlarged and placed next to the crack tip. Line 176
Page 6, line 197, what happens at -1200?
Page 7 Line 205-208. This part must be explained better. To study the steady/non-steady electrochemical process in the crack tip?
Page 7 line 208 –you mean crack tip? (should be a crack wall?)
Page 7 line 214 “the applied potential scan can be divided into the following three regions” Report these regions on the plots in fig. 6. It will help the reader to follow the discussion.
Page 7, line 216 tip was faster than the surface of the “metal”---should be crack wall
page 7 line 220-222 As shown in fig 7b?
Page 7 line 224, once you write non-crack tip, once surface of the metal.
Page 7, line 233 As shown in fig 7?
Page 8, Figure 6 - reports the scan rates also in the plots
Page 8, Figure 7, title of Figure 7 (after 3 days - reports the scan rates also here).
Page 9, line 256, First sentence - this is only true for -850 mV. for the others not.
How many times have you repeated these experiments? The stand deviation must be reported in table 1.
Language mistakes, page 4 line 151(sterile…)
Reviewer 2 Report
To the authors,
concerning the manuscript entitled “Effect of SRB and applied potential on stress corrosion behavior of X80 steel in high pH soil simulated solution” (materials - 1455500) by XU Cong-min, GAO Hao-ran, ZHU Wen-sheng, WANG Wen-yuan , SUN Can, Chen Yue-qing and XU Cong-min
Q1: Please remove unnecessary abbreviations from the abstract, especially if they are not defined in the context. Leave them to the introductory section.
Q2: Please re-write the introduction section in a way that the research is anchored in the field, the purpose is well defined and the degree of novelty is emphasized. Perhaps, the authors should assign one paragraph in which the overall aspects related to the corrosion of stainless steel is described, including the ways to circumvent that. Note that not all readers have experience in the field.
Q3: Please avoid constructions that rely on terms such as “synergy”. They hinder the cause-action principle in the first place. Please explain each contribution in part to pinpoint their specific influence.
Q4: Please define the SCC susceptibility, in Eq. 1. Also, please explain the values taken by SCC and how they translate in terms of corrosion resilience.
Q5: There is no impedance spectroscopy data in the manuscript. Please remove the information associated with that in the experimental description section.
Q6: The corrosion experiments are taken over very high potentials. The curves are complex and yet, not much is discussed except for the corrosion potential and current values. Please reserve the space for determining the parameters associated with the corrosion domains. There is a lot more information on corrosion resilience beyond the E/I corrosion values.
Q7: Chemical reactions are given and some arguments are added in the discussion section. Since the information is vital to picture the chemistry involved in corrosion experiments, one should move it up, to introduce the reader in the context.
Q8: One should re-organize the conclusions. I am not sure that the format used by the authors is in line with the journal format.
There are many passages that are difficult to follow. Also, terms such as “surface crack tip”, “non-crack tip” or “sterilesterile environment”, are ambiguous. Although the research shows scientific potential and novelty, the manuscript requires extensive editing and language check.
Reviewer 3 Report
In this manuscript, the authors present study of effect of SRB and applied potential on stress corrosion behavior of X80 steel in high pH soil simulated solution. Effect of SRB and applied potential on stress corrosion sensitivity of X80 pipeline steel was analyzed in high pH soil simulated solution under different conditions by slow strain rate tensile test (SSRT), electrochemical test and electronic microanalysis. Experimental results showed that X80 pipeline steel has a certain degree of SCC sensitivity in high pH simulated solution, and the crack growth mode belonged to trans-granular stress corrosion cracking (TGSCC).
This research is of great use in the field. Overall, the whole article is a good-written one with consecutiveness, strict logic, affluent datum, and clear consecution. The following suggestions are supplied:
- Please add some relevant simulation works regarding modeling the moving boundary problems/Stefan problems, which could arise during the corrosion, such as:
Michael Böhm et al., On a moving-boundary system modeling corrosion in sewer pipes, Applied Mathematics and Computation, Volume 92, Issue 2-3, 1998, pp 247–269.
- M. Ivanovic, et al., Numerical solution of Stefan problem with variable space grid method based on mixed finite element/finite difference approach, International Journal of Numerical Methods for Heat and Fluid Flow, 27, No. 12, 2017, pp. 2682-2695.
- Check English language in order to eventually improve it a little bit.
So, the present manuscript is suitable for publication in Materials, subject to the above mentioned minor revision points.
Author Response
I have modified it as suggested
Round 2
Reviewer 1 Report
The authors have sucessfully answered all questions and paper can be accepted.
Reviewer 2 Report
Dear authors,
here are my final suggestions for your manuscript materials-1455500
1) please correct "Sulfate Reudcing Bacteria (SRB)" in the abstract; Note that SRB is not defined in the abstract (in line 11)
2) the scales in Fig. 5 (insets) are hard to see;
3) in lines 303-305, the paragraph: "Authors should discuss the results ... directions may also be highlighted" does not bring more knowledge in the context. Perhaps, it should be discarded.